# A Bipolar-Channel Hybrid Brain-Computer Interface System for Home Automation Control Utilizing Steady-State Visually Evoked Potential and Eye-Blink Signals

**DOI:** 10.3390/s20195474

**Published:** 2020-09-24

**Authors:** Dalin Yang, Trung-Hau Nguyen, Wan-Young Chung

**Affiliations:** Department of Electronic Engineering, Pukyong National University, Busan 48513, Korea; dalinyang@pukyong.ac.kr (D.Y.); haunguyen85@pukyong.ac.kr (T.-H.N.)

**Keywords:** hybrid brain-computer interface (BCI), home automation, electroencephalogram (EEG), steady-state visually evoked potential (SSVEP), eye blink, short-time Fourier transform (STFT), convolution neural network (CNN)

## Abstract

The goal of this study was to develop and validate a hybrid brain-computer interface (BCI) system for home automation control. Over the past decade, BCIs represent a promising possibility in the field of medical (e.g., neuronal rehabilitation), educational, mind reading, and remote communication. However, BCI is still difficult to use in daily life because of the challenges of the unfriendly head device, lower classification accuracy, high cost, and complex operation. In this study, we propose a hybrid BCI system for home automation control with two brain signals acquiring electrodes and simple tasks, which only requires the subject to focus on the stimulus and eye blink. The stimulus is utilized to select commands by generating steady-state visually evoked potential (SSVEP). The single eye blinks (i.e., confirm the selection) and double eye blinks (i.e., deny and re-selection) are employed to calibrate the SSVEP command. Besides that, the short-time Fourier transform and convolution neural network algorithms are utilized for feature extraction and classification, respectively. The results show that the proposed system could provide 38 control commands with a 2 s time window and a good accuracy (i.e., 96.92%) using one bipolar electroencephalogram (EEG) channel. This work presents a novel BCI approach for the home automation application based on SSVEP and eye blink signals, which could be useful for the disabled. In addition, the provided strategy of this study—a friendly channel configuration (i.e., one bipolar EEG channel), high accuracy, multiple commands, and short response time—might also offer a reference for the other BCI controlled applications.

## 1. Introduction

A brain-computer interface (BCI) is a connection between a brain and a device that enables signals from the brain to direct various external activities without the participant of the peripheral nerve and muscles [1]. BCI is typically utilized by people with severe motor disabilities, such as patients with amyotrophic lateral sclerosis, brainstem strokes, or other neuromuscular diseases [2,3,4]. People can utilize BCI-based applications to control wheelchairs, prosthetics, toys, video games, and various computer applications. Additionally, a BCI-based home automation control system was recently proposed based on the promising advantage in the field of artificial intelligence. In contrast to other types of home automation systems (e.g., gesture and voice recognition), BCI-controlled home automation systems have no limitations in terms of lighting and noise conditions [5]. Also, it essentially allows a home automation system to be controlled based on user intentions directly.

To avoid the surgical procedures, there are four popular non-invasive measurement methods for acquiring the brain information: functional magnetic resonance imaging (fMRI) [6], magnetoencephalography (MEG) [7], electroencephalogram (EEG) [8], and functional near-infrared spectroscopy (fNIRS) [9,10]. Due to the advantage of the good temporal resolution, portable, and low cost, EEG shows unique usability superiority for the BCI-based control system in comparison to the other type of brain techniques (i.e., MEG, fMRI, and fNIRS) [11].

Traditionally, the EEG-based BCI system divided four various patterns, such as motor imagery (MI), Steady-state visual evoked potential (SSVEP), P300 potentials, and slow cortical potentials. Each category has its advantages and disadvantages [12]. One of novel literature [13] employed MI signals to control a system based on hand grasps, which cover only a small range of commands. Additionally, the MI model requires much training and suffers from significant accuracy problems due to the BCI illiteracy, which is sourced from the sensorimotor rhythms [14]. One of the research team [15] utilized eye blinks and eye movement as a control mode for a home lighting system and determined that eye blinking is more accurate compared to eye movement, which has a margin of error that may lead to unreliable results. In the comparison of MI and eye-blink signals, P300 and SSVEP signals showed an excellent performance (i.e., accuracy) and fast response times, which is the reason that most existing BCI-based home automation systems employ P300 [16,17,18,19,20,21], alpha rhythm [22,23], and SSVEP [24,25,26] signals for obtaining a faster performance and more accurate control. As the principle of P300 and SSVEP, both signals are based on external stimuli. P300 signals are excellent for multi-stimuli recognition (more than six), whereas SSVEP signals provide superior performance when the number of stimuli is less than six. Additionally, one reference [27] demonstrated that SSVEP signals yield faster responses to user mental activity and are less reliant on channel selection. The current study describes a hybrid BCI system that combines two/more than two brain techniques to control the external device, which could make up for the disadvantages of each individual technique. A review article [28] proved that hybrid BCI systems could provide more commands and provide the potential to increases the classification accuracy and information transfer rates. It was also determined by recent literature [29,30] on the hybrid BCI system, which used EEG (i.e., SSVEP and MI) and eye blink/movement signals as the input for a speller.

Most existing BCI-based home automation systems employ multiple channels to acquire EEG signals. However, multichannel data processing leads to considerable time consumption, and more electrodes cause higher costs and more complex experimental setups [31]. Also, the multiple channels configuration is the biggest challenge for daily usage, especially for disabled patients. Therefore, the selection of the proper EEG channels and the related brain cortex is significant. For SSVEP signal acquisition, one study [32] demonstrated that even though one can detect SSVEP signals from the scalp from non-hair-bearing regions, the occipital region provides satisfactory SSVEP signals. Also, a review article [33] states that the bipolar channel, based on the occipital region, can further enhance the signal to noise ratio of SSVEP. Additionally, eye closure induces a strong alpha wave, which can be detected via EEG or magnetoencephalography from the occipital region [34]. The above results reveal that both SSVEP and eye blink signals could be acquired from the occipital region, which further reduces the number of channels required for recording signals.

Among the studies published by BCI-based home automation researchers, many have been published based on multi-channel systems with numbers of commands ranging from 2–113. A summary of related studies is provided in Table 1. The highest accuracy of 94.17% was achieved by Goel [35] with a response time of 5.2 s to produce two commands using four EEG channels. As we mentioned above, two commands are not sufficient for system control in daily life. A more friendly system is essential with a lower number of channels configuration, a high classification accuracy and multiple control commands needs to be proposed.

This paper proposes a hybrid BCI-based home automation system utilizing SSVEP and eye blink signals to provide 38 commands (i.e., 6 × 6 SSVEP commands and two eye blink commands) for controlling daily life activities through a single bipolar channel. SSVEP signals provide selection functions, and single eye blinks provide the functionality to confirm selections. Resetting a selection requires one to perform a simple double eye blink.

The short-time Fourier transform (STFT) is applied to extract the feature. Moreover, the classification was conducted by a convolutional neural network (CNN). The offline/real-time results demonstrate that the proposed system could be used in daily life for home automation control with a robust classification accuracy and simple EEG headset structure by performing an easy task. The proposed system provides a novel strategy for a BCI controlled system. Also, this BCI application could offer the possibility for the disable people to utilize the home facility conveniently.

The rest of this paper is organized as follows: The materials and methodology regarding the brain signal based home automation system are introduced in Section 2, which includes the information of participants, device parameters, experimental protocol, and theoretical algorithm of brain signal processing. Section 3 illustrates the results regarding the optimal channel selection, performance comparison of different time windows, offline classification, and real-time evaluation. In Section 4, the proposed system is compared and discussed. Conclusions are presented in the last section.

## 2. Materials and Methods

### 2.1. System Architecture and Parameters

This study describes a hybrid BCI-based home automation system utilizing SSVEP signals and eye blinks. As shown in Figure 1, the proposed system consists of a SSVEP stimulus panel, BCI module, and visual home automation interface. The stimulus panel (Samsung, Seoul, Korea, 21.5”, 60 Hz refresh rate, 1920 × 1080 screen resolution) is utilized as the stimulus source. Each stimulus is a square with a side length of 4 cm, horizontal spacing between squares of 16.5 cm, and vertical spacing between squares of 9 cm. Six targets are presented in the BCI system with flicking frequencies of 6.6, 7.5, 8.57, 10, 11 and 12 Hz. The interface was designed in the C# platform, as shown in Figure 2, in which the text indicators were displayed along the flickers to make corresponding control by the user. A high-performance EEG device (Cognionics Inc., San Diego, CA, USA) was utilized to acquire eye blinking and SSVEP signals from the O-bipolar channel (i.e., O1 and O2) with the reference of the international 10–20 EEG system. HD-72 dry wireless EEG headset (Cognionics Inc., San Diego, CA, USA) is a commercial high-density EEG recording device, which contains the 64 EEG electrodes plus reference and ground. To utilize the low analog amplification and elimination of ac-coupling of the signal path, the 24-bit ADC is applied in the headset system. Furthermore, EEG signals are referenced to the right earlobe. The impedance of all electrodes was kept below 5 kΩ. The EEG data were amplified and digitalized with a sampling frequency of 500 Hz and a band-pass filter in the range of 1–50 Hz.

In this study, five healthy subjects (two females, three males, the median age of 24 years) with no prior brain-related or health issues participated after giving informed consent. The experiments were carried out following rules of the Declaration of Helsinki of 1975, revised in 2008. The identification code of approval is 1041386-202003-HR-11-02, approved on 3 March 2020, by the ethic committee of Pukyong National University. The subjects were asked to sit approximately 50 cm away from the monitor. They were then asked to focus their eyes on one of the six stimulus targets with flicking by the different frequencies (i.e., 6.6, 7.5, 8.57, 10, 11 and 12 Hz). EEG data were collected for 2 s for target identification, and a pop-up window would show the identification result (e.g., “Do you want to enter the sub-1 system?”). If the system detected a single-eye-blink signal, it meant the subject confirmed their selection. If a double-eye-blink signal was received, the system determined that the identified command was different from the subject’s intention. The system would then return to the previous interface and resume gaze control for the stimulus targets. After choosing the correct command, the participants could then further control the home automation system. A flow chart for this process is presented in Figure 3.

### 2.2. Setting Up the Interface

The interface of the proposed BCI-based home automation system was set up based on a 6 × 6 categorical system. A user first chooses one of the six main categories. The categories were designed based on the most common daily life activities, which could make life easier and more comfortable for users. The main categories are presented in Figure 2 with six categories of daily life control, calling, food ordering, conversation control, wheelchair control, and entertainment. Each category is further divided into six subcategories, which contain the common tasks relevant to each subcategory. Users can select an option by gazing at a flicker and confirm one selection via the single eye blinking. To undo a selection, a double blink will return to the previous menu. The interface was constructed in a C#-based visual environment. Each selection result is presented to users through pop-up windows. During the experiment, all six stimulus buttons were displayed simultaneously. The users were instructed to select their menu options by shifting their gaze to the corresponding button.

### 2.3. Experimental Protocol

*Step 1*: Setting up the headset on the user’s head and ensure that the scalp and electrode have good contact with good signal quality. The real-time data acquisition software (Cognionics Inc., San Diego, CA, USA) with its interaction channel between it and C# programming environment. As a result, EEG signals can be captured via Bluetooth communication using a PC’s serial port.

*Step 2*: Initiate the interface for the home automation system utilizing the Microsoft visual studio. A screen will appear on the monitor showing an interface with six main categories. To release the buffer pool of C#, once the buffer pool of C# cached more than 500 data samples of each channel, packaged MATLAB code was called for saving the EEG signals as the. mat file. After receiving two packages (i.e., 1000 data points), further analysis would be performed in MATLAB for generating the commands.

*Step 3*: To choose one of the six main categories, the user gazes at the corresponding category block. Each block is flickering with a different frequency (i.e., 6.6, 7.5, 8.57, 10, 11 and 12 Hz). The selected target would be encoded by the signal acquired from the occipital cortex. The subject then blinks their eyes to enter the subcategory menu (target block).

*Step 4*: A pop-up window will appear and ask the user if the window shows the correct selection. The user blinks their eyes again to confirm or blinks twice to return to the previous menu.

*Step 5*: A new window will appear, displaying the common tasks relevant to the selected category. The user can choose one term by blinking their eyes. Again, a pop-up window will appear to confirm the selection. They are blinking the eyes once will confirm the selection. Blinking twice will return to the previous menu.

### 2.4. Feature Extraction Protocol

A broad range of features has been implemented with the continuous development of BCIs to design BCI applications, such as the amplitude of EEG signals, band power, power spectral density, autoregressive models, and time-frequency features. In order to obtain effective output for BCI classification, it is necessary to understand clearly which features are accessible and how they are used. It is essential to select the relevant feature as the input for the classification [8]. As the literature [36] demonstrated that most brain activity patterns utilized to drive BCIs are defined by specific EEG time point and frequency band. Therefore, the time window of EEG signals should be considered as the important parameters during feature extraction Additionally, as the real-time analysis result indicated [37] that the EEG control range could facilitate to discover the beginning of alpha wave synchronization with low counts of false positives. Therefore, this study utilized the short-time Fourier transform (STFT) to simultaneously extract the features of the SSVEP signal and the eye-blink signal, which could contain the information in the time series and frequency band:(1)$$S\left(f,k\right)=\overset{N-1}{\underset{n=0}{\sum }}S\left(n\right)\left[w\left(n-k\right)e^{-\frac{j2\pi fn}{N}}\right].$$

After recording the data from the selected EEG channel, the infinite pulse filter is applied with a cutting-off frequency ranging from 5–30 Hz. 2 s EEG signals after the onset of the task was selected for further analysis. The EEG power was determined by the STFT algorithm utilizing functions of *spectrogram* (MATLAB™) over a 1 s (i.e., 500 data point) *Kaiser* window. The length of each step is 1, and the overlap window length is set up as a value of 499. The detailed calculation is shown in Equation (1), where *S*(*n*) is the original data in the time series, *ƒ* is the frequency, the window function is represented by *W*(*n*), *k* refers to the power. All features are derived from the time windows, and the oldest signal is eliminated from the active buffer when the new time series data reach. Then, the extracted feature is saved for the classification step. The procedure for signal processing is presented in Figure 4.

### 2.5. Classification

Convolution neural networks (CNN) can be used efficiently for the identification of characters and produce outstanding outcomes for multiple datasets [38], such as the MNIST database. A CNN model can accommodate geometric deformation, and the receptive field/convolutional kernel can be readily understood, and the forms of high-level features to identify are detected [39]. Therefore, numerous studies [40,41,42] have employed CNNs as classifiers to identify EEG signals. Network topology is the crucial feature in a CNN algorithm. Our Network topology is shown in Figure 4. Our network is made up of four layers with one or more maps in each. The CNN model measures the forwarding propagation activation by using a rectified linear unit as an activation function:(2)$$Z\left(u,v\right)=\sum_{i=-\infty }^{\infty }\sum_{j=-\infty }^{\infty }s\left(f,k\right)\cdot N1\cdot R\left(i,j\right)+\beta ,$$
(3)$$R\left(i,j\right)=\{\begin{array}{l} 1,\;\;j<2,0<i;\\ 0,\;\;Others; \end{array},$$
(4)a(u,v)=max(z(u,v)).

The normalization was performed for the extracted features with the frequency range of 5 to 30 Hz, which maintains the important information for the identification of the different features. The matrix of each input sample is 25 × 500. In this study, the size of the convolutional kernel (i.e., *N*) is 2 × 2. The bias is *β*. The output (i.e., *Z*) of the convolutional layer is calculated, as shown in Equation (2). Since the superiority of the fast speed of convergent, we applied the *Relu* function as the active function (i.e., shown in Equation (4)):(5)$$p\left(u,v\right)=w\cdot \overset{\infty }{\underset{i=-\infty }{\sum }}\overset{\infty }{\underset{j=-\infty }{\sum }}a\left(u,v\right)+\beta .$$

The pooling layer was used to reduce the size of the feature map. In this study, the max-pooling was conducted, which was employed to avoid the overfitting issue. In Equation (5), the weight is *W*, *P* represents the output of the convolutional layer. There were two fully connected layers (Layer 3 and Layer 4). This study employed the backpropagation to calculate the error term and gradient loss. The cost function is shown in Equation (6), as the input is given by Equation (7). Here, *h_w,b_*(*^(i)^*) is the desired values, and *y^(i)^* is the output value after the four-layers propagation:(6)$$J\left(w,b\right)=\frac{1}{m}\overset{m}{\underset{m=1}{\sum }}J\left(w,b;x^{\left(i\right)},y^{\left(i\right)}\right),$$
(7)$$J\left(w,b;x^{\left(i\right)},y^{\left(i\right)}\right)=\frac{1}{2}{\left(y^{\left(i\right)}-h_{w,b}\left(x^{\left(i\right)}\right)\right)}^{2},$$

After the calculation of each epoch, the unknown terms (i.e., *w b*) was updated with the negative lag direction. The algorithm is given by Equation (8) and Equation (9), respectively. The parameter α is the learning rate. After testing the trained CNN model, the error rate (ε = wrongly classified samples/total samples) for the testing sample was computed. The accuracy is calculated based on the equation: *accuracy* = (1 − *ε*) × 100%. Information transfer rate (ITR) is widely used om BCI filed [43], and it was calculated using the equations below:(8)$$w_{\left(i,j\right)}^{\left(l\right)}=w_{\left(i,j\right)}^{\left(l\right)}-\alpha \frac{\partial J}{\partial w_{\left(i,j\right)}^{\left(l\right)}},$$
(9)$$b_{\left(i,j\right)}^{\left(l\right)}=b_{\left(i,j\right)}^{\left(l\right)}-\alpha \frac{\partial J}{\partial b_{\left(i,j\right)}^{\left(l\right)}},$$
where *P* is the probability of detecting correct commands (i.e., refer to the accuracy in this study), *N* is the number of the commands performed, and *T* (i.e., 2 s) is the time required to produce the number of commands.
(10)$$\psi =\left(Plog_{2}P+\left(1-P\right)log_{2}\left(\frac{1-P}{N-1}\right)+log_{2}N\right),$$
(11)$$ITR=\frac{No\;of\;commands\times \psi }{T}.$$

## 3. Results

### 3.1. Channel Selection

Applying a large number of EEG channels may result in noisy or redundant signals that degrade BCI performance and user convenience. As demonstrated in [21], the occipital region provides the best SSVEP signals. In this paper, we compared the performances of three occipital channels (i.e., O1, O2, and O-bipolar) and choose the best electrode for online testing. The subjects were asked to sit in front of the screen and execute each command in turn. To reduce the error caused by changes in the environment, we compared the data acquired from the same subjects and time windows. Each channel recorded 800 trials (i.e., five subjects × eight tasks × 20 trials) and then utilized the CNN algorithm for training the model (i.e., 80% × 800 = 640 samples). The results after the testing (i.e., 20% × 800 = 160 samples), as shown in Figure 5, reveal that the O-bipolar channel provides the best performance (average accuracy of 96.92%) for all the subjects among the three channels. And the performance of O1 and O2 are no significant differences.

### 3.2. Time Window Selection

To calibrate the tradeoff between performance and time window duration, we trained the CNN models utilizing four different time-windows (1, 2, 3 and 4 s). The subjects were then asked to execute tasks based on cues on the screen. We created different segmentations for each of the time windows. Each time window contained 800 samples as input data. As shown in Figure 6, the 2 s time window provided good accuracy at 96.92%. Moreover, the 3 s and 4 s windows provided better performance by 97.28% and 98.51%, respectively. Considering the importance of time windows for the control system, the 2 s window is the selected for our system, since the 2 s time window could achieve the satisfied classification results. The classification performance (i.e., accuracy and loss) of different time windows are shown in Figure 7.

### 3.3. Feature Extraction

As shown in Figure 8, the extracted features were calculated by the STFT in the time (i.e., 2 s) and frequency domain. Totally, 800 feature maps (i.e., five subjects × eight tasks × 20 trials) were obtained from five subjects. Since the SSVEP frequency in subcategory interfaces are consistent with the main interface flickers, only six SSVEP features and two eye blinking features were extracted to train the model. In other words, once the selected feature was trained in the CNN model, the CNN model would recognize a similar pattern either during the selection in the subcategory interface or the main category interface.

In Figure 8, the power bar is shown on the right side. The feature of single eye blink present in Figure 8a; two power peaks were observed when the subjects performed double eye blink tasks, as shown in Figure 8b. The powerband for the SSVEP tasks with specific frequency are shown in Figure 8c–h, respectively, which refer to the brain features caused by gazing the different frequency flickering.

### 3.4. Real-Time Evaluation

In the online section, each subject performs eight trials to test the proposed system. Each trial includes a selection from the main interface, confirmation via eye blinking, selecting a command from the subcategory interface, and confirmation of the second command in sequence. The targeted selection from 36 control commands was decided by the willingness of each participant. The results for three subjects in the online evaluation experiments are listed in Table 2. As shown in Table 2, the mean eye-blink time for a single blink was 1.3163 s, with the fastest time for a single blink recorded for subject S1. The shortest time for sending a control command was 1.2 s and the longest time was 1.425 s. For a double blink, the longest time was 1.652 s, and the shortest time was 1.576 s, with a mean value of 1.608 s for sending control commands. In addition, the correct identified command of the single eye blink case was 47/48. In the double blink case, the five commands were detected correctly. In comparison with the eye blink, the SSVEP shows the lower ability (43/48) for identification.

## 4. Discussion

In this study, we designed a control mechanism for a BCI-based home automation system. The proposed system can identify 38 commands (i.e., 36 control commands and two calibration commands) utilizing single eye blinks, double eye blinks, and SSVEP signals recorded from a single bipolar channel with a classification accuracy of 96.92%. As the best of the authors’ knowledge, this is the first study that utilizes only one bipolar EEG channel for a home automation control BCI system with good accuracy within a short time window. The proposed system could provide a novel way for physical disordered people to control external devices by gazing and eye blinks. It offers a possibility to conduct daily routine tasks using brain signals directly without any physical movement.

In comparison with the previous relevant research [29,30,44,45,46,47] (listed in Table 3), instead of using the eye-tracking to detect the eye movement, this study applied the eye blinking to be a trigger in this control system. In the eye-tracking system, the extra device is required to monitor the eye movement. As demonstrated in the hybrid eye-tracking and SSVEP system [30], the participants need to wear the extra video eye-tracking system, for which the threshold of velocity, acceleration, and minimum deflection was 30°/s, 8000°/s^2^, and 0.1°, respectively. Since the eyeblink and the SSVEP signal can be acquired from the EEG device with the same channel, the unfriendly hardware burden and extra cost of the hybrid device (e.g., EEG and eye-tracking device) will be reduced. With the development of the eye-tracking, the pure eye-tracking system is going to be an alternative technique of the BCI. However, one reference reported that the EEG-based system is easiest to use, also, the SSVEP-based system shows better performance than the eye-tracking system [48].

Five subjects participated in an offline training experiment to evaluate the performance of different channels and time windows. The results demonstrate that the O-bipolar channel provides better performance compared to channels O1 and O2, as shown in Figure 5. Based on our results, we concluded that utilizing the O-bipolar channel significantly reduces the interference of noise. Response time plays a vital role in real-time systems. Therefore, four different time windows for data partitioning are utilized for the evaluation in this study. Figure 7 presents a comparison performance (i.e., classification accuracy and loss) between the different time windows. The 4 s time windows offer the best performance in terms of accuracy (i.e., 98.51%), as more information is included for the longer time window in comparison to the short time windows. This finding is consistent with the pioneer study [45], which could achieve 100% by conducting the task with a 20 s time window. With the development of deep learning, an improved classification may be achieved by utilizing the hybrid modality (i.e., EEG and fNIRS) [49,50] advanced machine learning algorithms, such as long-short team memory [51] and deep neural network [52]. In addition, in this study, we applied red squares with the text indicators to guide the participants to select the corresponding control commands. A more intuitive display method might provide a more friendly interface (e.g., pictures, or different color squares, or various shapes) for the user and we will consider this in our future work.

For the time windows selection, we applied four different time windows (i.e., 1, 2, 3, and 4 s) to assess their performance. As shown in Figure 6, the time window of 4 s could achieve the highest accuracy compared with the other time windows (e.g., 1, 2, and 3 s). However, to reduce the time consumed of this proposed control system, we choose 2 s as the time window to obtain a satisfactory result (i.e., 96.26%). In the online experiment, three subjects were recruited to control the proposed system in real-time. Each participant performed eight trials, which were conducted in sequence: (i) selection in the main interface, (ii) calibration via eye blinking, (iii) selection from the subcategory interface, and (iv) calibration for the second selection. Before the experiment, the participants were informed to select all the flickers in the main interface. The decision of selection in each subcategory interface was made by the subjects randomly. Thus, part of the commands from the 36 functions was evaluated. As shown in Table 2, the initial investigation results indicated the feasibility of the proposed system. In future work, we will develop our own simplified device (i.e., two electrodes) and examine the real-time system with a total of 36 functions.

Although the offline and real-time virtual results showed good performance (i.e., single bipolar channel, good accuracy, and short time window) for the proposed home automation control system, some limitations need to be mentioned. First, the system is simulated in the virtual environment. The real home automation application results may lead to less accuracy due to the lousy human mental state (e.g., distracted and motor artifact, etc.) and the signal transmission problem between the home automation application and the EEG device. Secondly, eye blink is typically considered an undesirable electrical potential. With the advantage of high amplitude and analytic features, voluntary eye blinks are widely employed as an input or control command in BCI areas. Therefore, one needs to pay attention during the analysis to noise reduction. Also, the control of the eye blink should follow the cue while performing the task. In addition, the idle state/resting state was not considered for state identification. Non-detection of the idle state may lead to misclassification during long duration experiments. For a future study it was suggested to add an extra idle state detection to avoid this issue. Lastly, the objective of this study was to investigate the feasibility of home automation control application with a single bipolar channel by using the hybrid SSVEP and eye blink signals. In this pilot study, only a few participants were considered (i.e., a total of 800 trials was conducted to increase the dataset of the CNN classification subjects to assess the repeatability and stability). In future work, more participants need to be investigated for comprehensive analysis in a real-time environment.

## 5. Conclusions

This study proposed a hybrid BCI-based home automation system utilizing SSVEP and eye-blink signals with a single bipolar channel for multiple comment control (i.e., 38 commands). SSVEP signals are utilized to select desired commands, and eye-blink signals are utilized to calibrate command selections. Both signals are obtained from the same bipolar channel and classified by the same CNN model. Our experiments included two modules for processing and analyzing EEG signals. An offline module was employed to assess general model performance (e.g., channel selection, time window selection, feature extraction, and CNN model training). Five subjects participated in offline experiments. The results demonstrated that 38 daily task commands could be identified with an accuracy of 96.92% based on a 2 s time-window using the signal acquired from the O-bipolar channel. Three subjects participated in a real-time experiment, and the results demonstrated that changes in brain intentions could automatically control the proposed system. As the best knowledge of the authors, this is the first work to utilize the combination of SSVEP and eye-blinks to perform home automation control. This study demonstrated that it is possible to achieve multidimensional control with good performance using SSVEP and eye blink signals from only one single bipolar channel. Also, the proposed system could be applied to home automation control system, which could be a helpful assistant for disabled/healthy people.

## Figures and Tables

**Figure 1 sensors-20-05474-f001:**
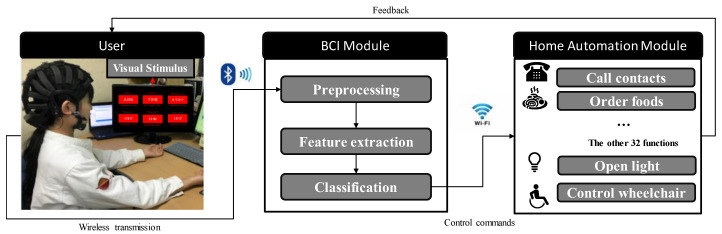
The control structure for our hybrid BCI system.

**Figure 2 sensors-20-05474-f002:**
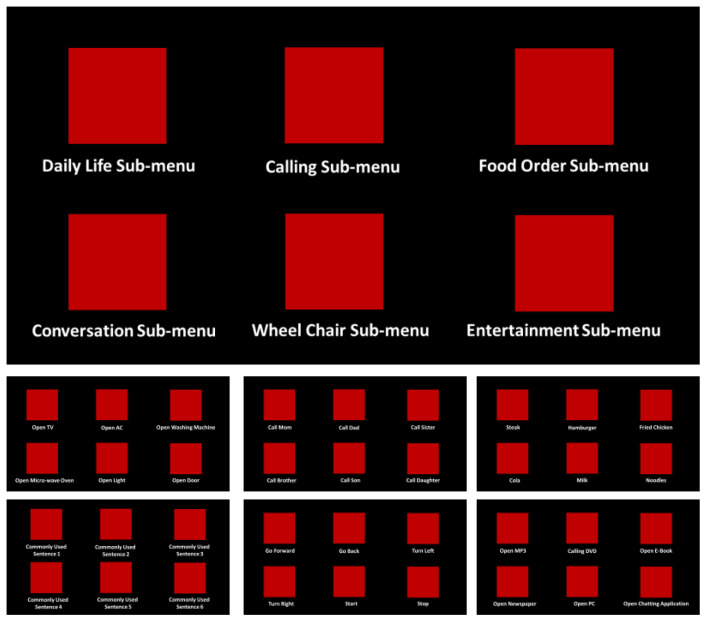
Interface for the selection menu.

**Figure 3 sensors-20-05474-f003:**
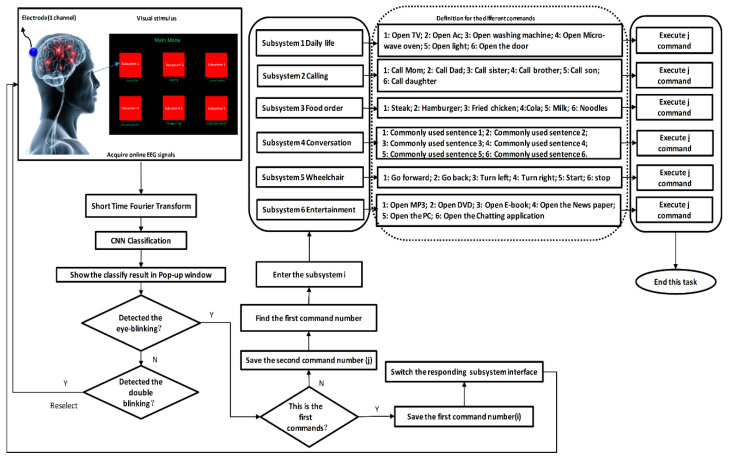
Control strategy for our hybrid BCI-based home automation system.

**Figure 4 sensors-20-05474-f004:**
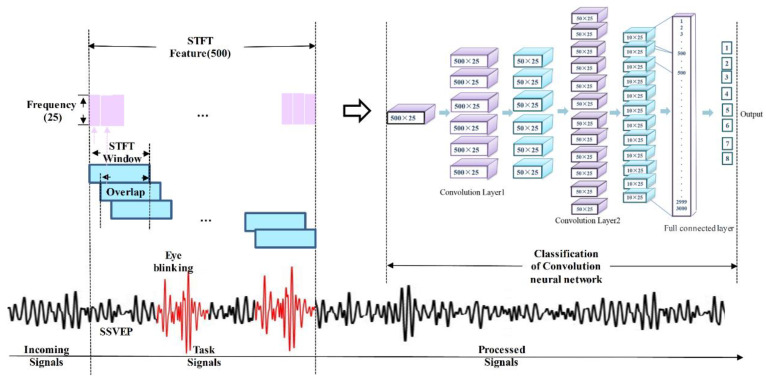
Procedure for signal acquisition, feature extraction, and classification.

**Figure 5 sensors-20-05474-f005:**
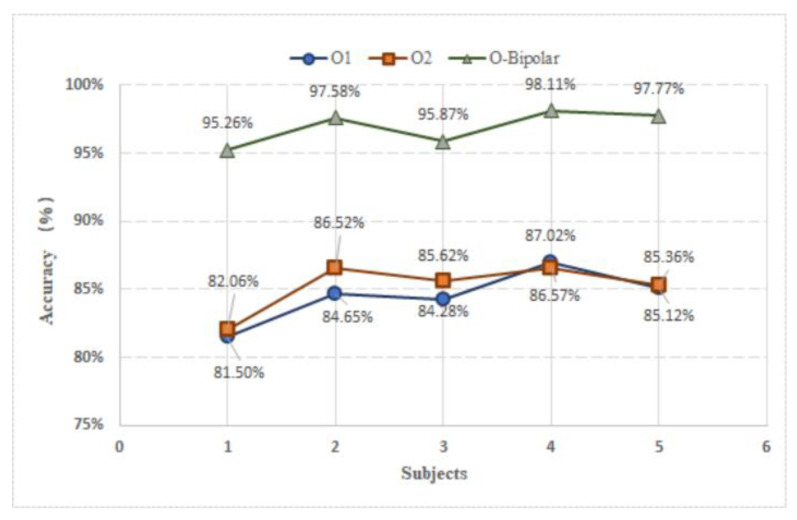
Performance comparison between different channels.

**Figure 6 sensors-20-05474-f006:**
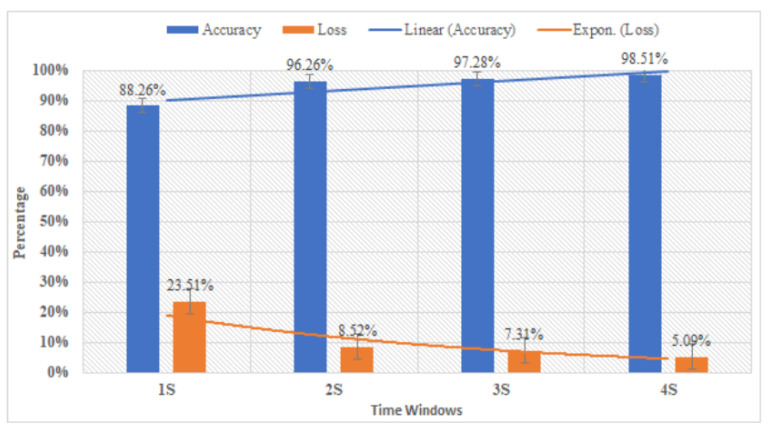
Performance comparison between different time windows.

**Figure 7 sensors-20-05474-f007:**
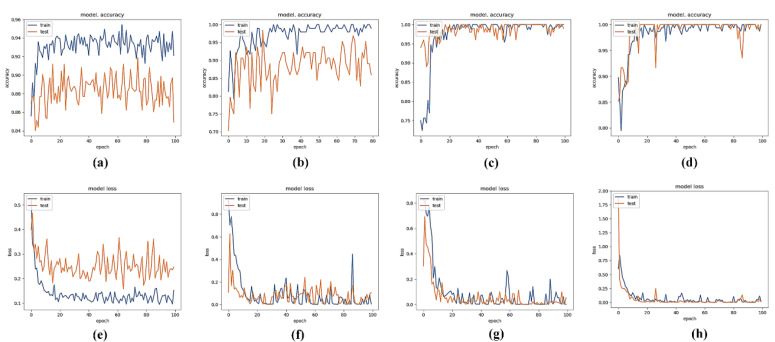
Performance results for different time windows, the accuracy results for the 1 s, 2 s, 3 s, and 4 s time windows are shown in (**a**–**d**), respectively. The corresponding training losses are shown in (**e**–**h**).

**Figure 8 sensors-20-05474-f008:**
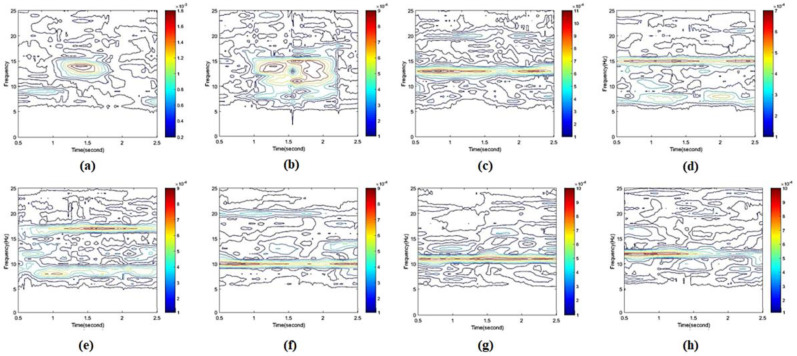
STFT features for (**a**) single eye blink, (**b**) double eye blink, (**c**) SSVEP at 6.6 Hz, (**d**) SSVEP at 7.5 Hz, (**e**) SSVEP at 8.57 Hz, (**f**) SSVEP at 10 Hz, (**g**) SSVEP at 11 Hz, and (**h**) SSVEP at 12 Hz.

**Table 1 sensors-20-05474-t001:** Summary of related works on BCI-based home automation systems.

Publication	Type	Category	Commands	Channel	Evaluation Criteria		
					Accuracy (%)	Time (s)	ITR (bits/min)
Aloise et al. [17]	Real	P-300	16	8	90.00	4–5.6	11.19
Holzner et al. [21]	Virtual	P-300	13	N/A	79.35	N/A	N/A
Karmali et al. [23]	Virtual	Alpha Rhythm	44	4	N/A	27.7	N/A
Kosmyna et al. [5]	Virtual	Conceptual imagery	8	16	77–81	N/A	N/A
Goel et al. [35]	Virtual	SSVEP + Eye blink	2	4	94.17	5.2	11.6
Lin et al. [22]	Virtual	Alpha Rhythm	2	1	81.40	36–37	N/A
Prateek et al. [25]	Virtual	SSVEP	5	8	84.80	15	N/A
Perego et al. [26]	Virtual	SSVEP	4	N/A	N/A	350	N/A
Corralejo et al. [20]	Virtual	P-300	113	8	75–95	10.2	20.1
Carabalona et al. [19]	Real	P-300	36	12	50–80	N/A	N/A
Sellers et al. [18]	Real	P-300	72	8	83.00	N/A	N/A
Our study	Virtual	SSVEP + Eye blink	38	1	96.92	2	146.67

**Table 2 sensors-20-05474-t002:** Real-time results for three subjects controlling the virtual home automation system.

Subjects	Correct Command			Response Time		
	N_eye-blink_	N_double-blink_	N_SSVEP_	T_eye-blink_ (s)	T_double-blink_ (s)	T_SSVEP_ (s)
S1	16/16	1/2	14/16	1.25	1.652	1.754
S2	15/16	2/2	15/16	1.425	1.576	1.953
S3	16/16	2/2	14/16	1.274	1.597	1.731
Total	47/48	5/6	43/48	1.316	1.608	1.813
Std.	0.036	0.289	0.036	0.095	0.039	0.122

N: Number; T: Time; Std: Standard Deviation.

**Table 3 sensors-20-05474-t003:** Related works regarding the hybrid EEG and eye blink BCI system.

Study	Type	Category	Commands	Channel	Accuracy (%)	Time (s)	ITR (bits/min)
Wang et al. [44]	Real	P-300 + Eye blink + MI	8	17	91.25	4	N/A
Wang et al. [45]	Virtual	SSVEP + Eye blink	12	4	40–100	0–20	N/A
Park et al. [46]	Virtual	SSVEP + AR + Eye blink	16	32	92.8	20	37.4
Duan et al. [47]	Real	MI + SSVEP + Eye blink	6	12	86.5	48	1.69
Malik et al. [30]	Virtual	SSVEP + Eye tracking	48	8	90.35	1	184.06
He et al. [29]	Virtual	MI + Eye blink	N/A	32	96.02	6.16	45.97
Our study	Virtual	SSVEP + Eye blink	38	1	96.92	2	146.67

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
