# Peer review of "A Bipolar-Channel Hybrid Brain-Computer Interface System for Home Automation Control Utilizing Steady-State Visually Evoked Potential and Eye-Blink Signals"

_sensors, 2020, doi:10.3390/s20195474_

Round 1
Reviewer 1 Report
Please find the comments in the attachment.

Author Response
Authors would like to thank the Editor and reviewers for their valuable time to review and critique our manuscript. The manuscript has been thoroughly revised upon the reviewers’ comments. Authors’ point-by-point answers to the comments are provided below.
Comment 1: Recording system is not very well described: How many electrodes (pos, neg, gnd), amplification, filters (bandwidth/slope), ADC bits (only stating sampling time is not adequate).
Answers: The elaborated description of the recording system has been provided in the revised manuscript. The following sentence has been added.
“HD-72 dry wireless EEG headset is a commercial high-density EEG recording device, which contains the 64 EEG electrodes plus reference and ground. To utilize the low analog amplification and elimination of ac-coupling of the signal path, the 24-bit ADC is applied in the headset system.” Thank you!
Comment 2: System Training and Testing are little or poorly described. How was the training/testing procedure implemented for each subject? What happens when the subject does not intend anything (just idle between trials)? Are the training samples subject specific for each subject or general for all? The main goal of this study is to combine gaze and blink targets, so we need the training and testing results for each and population. How are the accuracies and losses defined and computed in detail? Referring to two popular equations (Eq. 10 and 11) do not add anything to the strength of the study. How is P is computed for each target and how are the T.P., TN, F.P., F.N. values are determined?
Answers: The reviewer’s concern is right. Indeed, the said details need further description and elaboration.
(1) The training procedure was implemented by the convolutional neural network in the offline mode, which is mentioned in subsection 3.1. And the testing procedure was divided into two categories: the offline testing was conducted by 20% of the samples for testing the performance of the trained CNN model, and the online testing was performed by the real-time system, as mentioned in subsection 2.3.
(2) In this proposed system, the authors did not consider the idle state/resting state. We discussed this limitation in the discussion section of the revised manuscript as below.
In addition, the idle state/resting state was not considered for state identification. Non-detection of the idle state may lead to misclassification during the long-period experiment. The future study was suggested to add the extra idle state detection for avoiding this issue.
(3) The training samples are generated from all the subjects. This study aims to investigate the feasibility of using a single bipolar EEG channel to achieve home automation control with satisfactory classification accuracy in a short time window. The training and testing one by one for each subject may lead to a higher computational cost and an increase in processing time. In this study, the one by one testing was only applied in the online experiment.
(4) After testing the trained model, the error rate (ε = wrongly classified samples/total samples) for each input sample were computed in comparison with true labels (known). Then, the accuracy was calculated as follows. The loss was computed by Equation (6) and (7) of the revised manuscript.
(5) The reason for using Equation (10) and (11) is to define the calculation of information transfer rate. In this study, the authors did not calculate the confuse matrix. Instead of the confusion matrix (i.e., T.P., TN, FP, and FN), the authors employ accuracy and loss as the index to quantitative the classification performance.
Author’s have now added all these details in the revised manuscript. Thank you for this constructive comment. Authors believe that it has truly helped strengthen the methodology and results section of the manuscript.
Comment 3: Results are given in four subsections: channel, time-window, feature extraction/classification, real-time evaluation. We should expect numerical results for each. We only get data for the first two, some for the third (Fig. 9 is missing; see line 247) and some for the last. Real-time testing results are given only for 3 subjects (Table 2) and only with respect to blinks. How about the 6 flickering choices detected by gazes? How about other subjects?
Answers:
(1) The authors revised the results section and elaborated on the numerical results for each subsection in the updated manuscript.
(2) Figure 9 was miswritten; in actual, it represents Figure 8. The authors have made appropriate changes in the manuscript.
(3) Real-time testing was confined to 3 subjects only to assess the possibility of a real-time system. Considering the long-range of subjects were included in our future work scope. The limitation of the subject in the real-time system has been discussed in the Discussion section.
(4) Flickering choices detected by gazes are shown in Table 2, in NSSVEP and TSSVEP columns, respectively.
All these details have been incorporated in the revised manuscript. Thank you!
Comment 4: Eye blinks are easily determined in frontal EEG, but they are not that obvious in occipital electrodes such as O1 and O2 used in this study. Since EEG recording montage is not described in detail, we are not sure how they are detected properly. Our only information comes from example from Fig.4. Is that an example of single blink or double? Blink detection (muscle EMG) and gaze detection (SSVEP) should be described more in detail
Answers:
As the literature [1] shown that the occipital lobe during the period eye blink (eye close & open) could generate a strong alpha wave (8-12 Hz). In order to minimize the number of channels, we just considered the occipital electrodes for recording the SSVEP and eye blink signal. The occipital cortex selection as the region to acquire the data has been explained in the Introduction section. And more explanations for the recording system have been added in the revised manuscript to avoid unnecessary misunderstanding. Figure 4 is a represents the general overall procedure for incoming signals until the processing of the signals. It covers both blinks as well as gaze detection. The detected features for the blink and gaze (SSVEP) are shown in Figure 8, which shows the difference between the various features. Thank you.
[1] Palva S. and Palva J.M. (2007). "New vistas for a-frequency band oscillations". Trends Neurosci. 30 (4): 150–158.
Comment 5: Unnecessary References The study contains too many of them. P300 gets 6 references [16-21] which is not used or needed in this study while SSVEP gets only 3 or 4 [24-25]+[27] (some in short proceedings articles). The same is true for fNIRs and others. They should be mentioned in one or two references. On the other hand, little review on SSVEP (flicker, checkerboard reversal etc) especially wrt gaze and visual field is provided. Eye blink recordings are very important especially for this study. They should be properly reviewed.
Answers: Thank you for your comment. The reason for using the six references regarding the P-300 was to descript the related work regarding the brain-computer interface-based home automation control system, as listed in Table 1. On the contrary, less literature is used in the home automation control system by applying the SSVEP and eye blink.
Comment 6: References: Not proper referencing: Use “et al. (year)” when referring an article with three or more authors; use both authors when there are two. 13: Zakaria et al.; 15: Rani and Wadidah; 27: Zhao et al. Please check all the manuscript.
Answers: The authors double-checked the referencing format to make it sure throughout the manuscript by following the journal guidelines.
Comment 7: Who is D.L. in author contributions (ln 317-318)? Should it be D.Y.?
Answers: Thank you for pointing out the mistake; we corrected the name as D.Y.
Comment 8: Put system or procedure descriptions in the Methods section not in the Results (see for example lines 241-243)
Answers: In compliance with the reviewer’s comment, the authors shift the procedure descriptions to the Method section.
Comment 9: Why is 2s is selected as the best window? “Most suitable” based on what? There should be some rational thinking on that. Does ”sending a control command” (lines 253-256) have anything to do with it?
Answers: As shown in Figure 6, there was not a significant improvement (i.e., accuracy and loss) for the 3s and 4s time window. Considering the importance of time windows for the control system, the 2 s window is the selected for our system, since the 2 s time window could achieve the satisfied classification results. In addition, the response time is not considered for the selection of time windows. The necessary explanation was provided in the revised manuscript for the clear description. Thank you.
Reviewer 2 Report
This paper proposes a hybrid brain-computer interface for home automation control. The following points are my main concerns.
- The motivations of using BCI for home automation control should be further emphasized. The limitations of using BCI for home automation control should also be discussed. Is conductive gel needed in this experiment? How do the authors propose to conquer the inconvenience of wearing the cap and applying conductive gel required by using BCI?
- The control accuracy and time of using BCI for home automation control should be compared to other methods for home automation control as well as the manual control method.
- How do the authors propose to achieve intuitive control by showing just red squares on the screen to represent different tasks?
- Has this study been approved by any ethical committee?
Author Response
Authors would like to thank the Editor and reviewers for their valuable time to review and critique our manuscript. The manuscript has been thoroughly revised upon the reviewers’ comments. Authors’ point-by-point answers to the comments are provided below.
Comment 1: The motivations of using BCI for home automation control should be further emphasized. The limitations of using BCI for home automation control should also be discussed. Is conductive gel needed in this experiment? How do the authors propose to conquer the inconvenience of wearing the cap and applying conductive gel required by using BCI?
Answers: Thank you for your comments.
(1) The motivation of using the BCI based home automation control system was further emphasized in the Discussion section (i.e., first paragraph). Meanwhile, the limitations of this study were also discussed in the Discussion section (i.e., first paragraph).
(2) In this study, the authors used dry electrodes during experiments. The details for the recoding system have been added in the Method section for a clear description.
Comment 2: The control accuracy and time of using BCI for home automation control should be compared to other methods for home automation control as well as the manual control method.
Answers: The authors discussed the related work in Table 1 and Table 3. In this study, the authors aimed to investigate the feasibility of using a single bipolar EEG channel to achieve home automation control with a satisfactory classification accuracy by a short time window to the physically disabled people. Therefore, the manual control system is not considered and compared in the manuscript. Thank you.
Comment 3: How do the authors propose to achieve intuitive control by showing just red squares on the screen to represent different tasks?
Answers: In this study, the red blocks are actually flickering with different frequencies (i.e., 6.6 Hz, 7.5 Hz, 8.57 Hz, 10 Hz, 11 Hz, and 12 Hz). The subjects can select by gazing at the target block to generate the SSVEP signals, which will be acquired by the electrodes, and later signal will later be encoded as the target frequency command. The authors have added the explanation for the target selection in the Method section of the revised manuscript.
Comment 4: Has this study been approved by any ethical committee?
Answers: Yes, the experiments were carried out under the approval of the ethics committee. The number of approval is 1041386-202003-HR-11-02.
Round 2
Reviewer 1 Report
please find the attachment

Author Response
The authors would like to thank the Editor and two anonymous reviewers for their valuable time to review and critique our manuscript. The manuscript has been thoroughly revised upon the reviewers’ comments. The authors’ point-by-point answers to the comments are provided below.
Comment 1: In this revision the authors have made many changes and improvements and they are commended for that. they are, however, not systemic and I still see many problems especially in the newly revised text in the use of English and grammar. Some references are still made without using et al. (for example, see Table 1: Karmali (23) or Wang (ln 76); both have many coauthors as observed in the references) or simple grammar problems (for example, ln252-253 (revised in red): “which are represent the brain features”) or very unnecessarily long, cumbersome sentences (ln79-82). I strongly recommend a thorough inspection/revision of the manuscript.
Answers: The authors are thankful for the encouraging remarks above and pointing out the mistakes. We have thoroughly examined the entire manuscript to remove typos or any grammatical mistakes and have corrected the references by following the journal guideline.
Comment 2: Still two major problem of the manuscript are the methods and testing. Both are seen in the Table 1 as the claim of 38 commands; 1 channel (2 electrodes); 96.92% accuracy; 2 s time; 146.67b/m ITR). This claim means real-time performance results and there is not enough material in the manuscript to fully support it.
(A) Is 1 of 38 commands fully classified in 2s? This is a 2-tier system which has to go first 6 commands (parallel) then the blink check and then go through another 6 command with another blink all in serial. Performance result of Fig.5 and related text sounds like only 8 targets are checked off-line. The authors need to specify how 36 targets were tested separately for all 5 subjects (sessions, trials, sequences, accuracy, idle times, time measurement, etc) and report the outcome. Otherwise it is not fair to the other authors listed in Table 1.
(B) Do we really have 2 electrodes (O1 and O2)? Why are we then using a HD-72 EEG headset with EEG referenced to the bilateral ear (ln118-123)? Where on the ear is the reference electrode (linked mastoid (requires 2 electrodes) or ear lobes?) Who makes this H D-72 headset?
Answers: The authors have added the explanation to support the claim mentioned in Table 1. As per review comments, the response for each question was answered as below, separately.
(A) The proposed system could identify 38 commands (i.e., 36 control commands and 2 calibrating commands) utilizing single eye blinks, double eye blinks, and SSVEP signals by a single bipolar channel. In compliance with the reviewer’s comments, the authors have added an explanation regarding the offline and online system in the revised manuscript. For your reference please find the additional description added in the revised manuscript as below:
Subsection 3.3: As shown in Figure 8, the extracted features were calculated by the STFT in the time (i.e., 2 s) and frequency domain. Totally, 800 feature maps (i.e., five subjects × eight tasks × 20 trials) were obtained from five subjects. Since the SSVEP frequency in subcategory interfaces are consistent with the main interface flickers, only six SSVEP features and two eye blinking features were extracted to train the model. In other words, once the selected feature was trained in the CNN model, the CNN model would recognize a similar pattern either during the selection in the subcategory interface or the main category interface.
Subsection 3.4: In the online section, each subject performs eight trials to test the proposed system. Each trial includes a selection from the main interface, confirmation via eye blinking, selecting a command from the subcategory interface, and confirmation of the second command in sequence. The targeted selection from 36 control commands was decided by the willingness of each participant.
Discussion: In the online experiment, three subjects were recruited to control the proposed system in real-time. Each participant performed eight trials, which were conducted in sequence: i) selection in the main interface, ii) calibration via eye blinking, iii) selection from the subcategory interface, and iv) calibration for the second selection. Before the experiment, the participants were informed to select all the flickers in the main interface. The decision of selection in each subcategory interface was made by the subjects randomly. Thus, part of the commands from the 36 functions was evaluated. As shown in Table 2, the initial investigation results indicated the feasibility of the proposed system. In future work, we will develop our own simplified device (i.e., two electrodes) and examine the real-time system with a total of 36 functions.
(B) The EEG signals were acquired from the O1 and O2 channels by setting the headset for only these two active electrodes. In our lab, we used the HD-72 dry EEG headset (Cognionics Inc., San Diego, USA) for multiple experiment paradigms by using different channel configuration, since this commercial device could obtain the robust and clear EEG signals, that is why we utilized this device to assess the feasibility of the proposed system. In future work, we will develop our own simplified device (i.e., two electrodes). The authors have mentioned the information regarding the channel configuration and headset in the revised manuscript.
Reviewer 2 Report
This paper looks better now. However, the authors still have not answered the question that just showing red squares with different flickering frequencies on the screen for the user to look at to control different tasks does not seem an intuitive control. It would be better, for example, if pictures representing the correspoinding tasks are shown along with the red squares. Or there may be some other ways to make this control more intuitive for the users. Discussion regarding this is advised to be added.
Moreover, there is no discussions regarding the limitations of this study in the first paragraph of the discussion section as the authors claimed. The advantages and disadvantages of the proposed method compared to pure eye tracking control should be further discusssed regarding the application of home device control for physically disordered people.
Please add the ethical approval information to the paper.
Author Response
The authors would like to thank the Editor and reviewers for their valuable time to review and critique our manuscript. The manuscript has been thoroughly revised upon the reviewers’ comments. The authors’ point-by-point answers to the comments are provided below.
Comment 1: This paper looks better now. However, the authors still have not answered the question that just showing red squares with different flickering frequencies on the screen for the user to look at to control different tasks does not seem an intuitive control. It would be better, for example, if pictures representing the corresponding tasks are shown along with the red squares. Or there may be some other ways to make this control more intuitive for the users. Discussion regarding this is advised to be added.
Answers: Thank you for the thoughtful comment. The authors misunderstood the reviewer’s comment in the last revision and appreciated the interesting ideas suggested by the reviewer. In this study, Only the text indicators were displayed along the flickers to make corresponding control by the user, as the demonstration shown in Figure 2. The authors have added the explanation in subsection 2.1 of the revised manuscript for easier understanding to the audience. As per the reviewer’s suggestion, we also discussed the ways for the various flickering interface in the updated manuscript. For your kind information, the added discussion was attached as below:
Discussion: In addition, in this study, we applied the red squares with the text indicators to guide the participants to select the corresponding control commands. A more intuitive display method may provide a more friendly interface (e. g., pictures, or different color squares, or various shapes) for the user and we will consider it in the future work.
Comment 2: Moreover, there is no discussions regarding the limitations of this study in the first paragraph of the discussion section as the authors claimed. The advantages and disadvantages of the proposed method compared to pure eye tracking control should be further discussed regarding the application of home device control for physically disordered people.
Answers: The authors apologize for the mistake of non-marking the revised part for the limitation of our study in the last revision. The limitation of this study was discussed in the last paragraph of the Discussion section of the updated version. As per the review’s comments, the authors further discussed the advantages and disadvantages of the proposed system compared with the pure eye-tracking control home automation system. For your kind reference, the details are listed as follows.
Discussion: In the eye-tracking system, the extra device is required to monitor the eye movement. As demonstrated in the hybrid eye-tracking and SSVEP system [30], the participants need to wear the extra video eye-tracking system, for which the threshold of velocity, acceleration, and minimum deflection were 30º/s, 8000 º/s2, and 0.1º, respectively. Since the eyeblink and the SSVEP signal can be acquired from the EEG device with the same channel, the unfriendly hardware burden and extra cost of the hybrid device (e.g., EEG and eye-tracking device) will be reduced. With the development of the eye-tracking, the pure eye-tracking system is going to be an alternative technique of the BCI. However, one of the literature reported that the EEG based system is most comfortable to use, also, the SSVEP based system shows better performance than the eye-tracking system [48].
Comment 3: Please add the ethical approval information to the paper.
Answers: The authors have added the ethical approval information in the updated manuscript in line 127 and line 130 on page 4. Thank you.